# Synthesis, Characterization, and Antibacterial Evaluation of a Cost-Effective Endodontic Sealer Based on Tricalcium Silicate-White Portland Cement

**DOI:** 10.3390/ma14020417

**Published:** 2021-01-15

**Authors:** Indra Primathena, Denny Nurdin, Hendra Hermawan, Arief Cahyanto

**Affiliations:** 1Department of Conservative Dentistry, Faculty of Dentistry, Universitas Padjadjaran, Bandung 40132, West Java, Indonesia; indra.primathena@fkg.unpad.ac.id (I.P.); denny.nurdin@fkg.unpad.ac.id (D.N.); 2Department of Mining, Metallurgical and Materials Engineering, Faculty of Sciences and Engineering, Université Laval, Quebec City, QC G1V0A6, Canada; hendra.hermawan@gmn.ulaval.ca; 3Medical Devices and Technology Centre (MEDiTEC), Institute Human Centred Engineering (iHumEn), Universiti Teknologi Malaysia, Johor Bahru 81310, Malaysia; 4Oral Biomaterials Study Centre, Faculty of Dentistry, Universitas Padjadjaran, Bandung 40132, West Java, Indonesia; 5Department of Dental Material Science and Technology, Faculty of Dentistry, Universitas Padjadjaran, Bandung 40132, West Java, Indonesia

**Keywords:** antibacterial, endodontic sealer, *Enterococcus faecalis*, mineral trioxide aggregate, tricalcium silicate, white Portland cement

## Abstract

Mineral trioxide aggregate (MTA) is an ideal yet costly endodontic sealer material. Tricalcium silicate-white Portland cement (TS-WPC) seems to have similar characteristics to those of MTA. This work aims to characterize a modified TS-WPC and evaluate its antibacterial properties as a potential endodontic sealer material. The modified TS-WPC was synthesized from a 4:1 mixture of sterilized Indocement TS-WPC and bismuth trioxide using a simple solution method with 99.9% isopropanol. The mixture was stirred until it was homogenous, centrifuged, and dried. The material was then characterized using infrared spectroscopy, X-ray diffraction, and electron microscopy and subjected to antibacterial evaluation against *Enterococcus faecalis* using a Mueller–Hinton agar inhibition test. The results showed that the material was characterized by main functional groups of hydroxyls, silicate, bismuth trioxide, and tricalcium silicate, like those of a commercial MTA-based sealer, both tested after hydration. Modified TS-WPC before hydration showed similar powder morphology and size to the commercial one, indicating the ease of manipulation. Both materials exhibited antibacterial activity due to calcium dihydroxide’s ability to absorb carbon dioxide, which is essential for the anaerobic *E. faecalis*, with minimum inhibitory effect and bactericidal concentrations of 12,500 ppm and 25,000 ppm, respectively. The modified TS-WPC has the potential to become a cost-effective alternative endodontic sealer material.

## 1. Introduction

Endodontic treatment is regularly performed to eliminate necrotic tissue and pathogenic bacteria and assist the teeth healing process surrounding the damaged tissue while preventing bacterial reinfection. The treatment consists of three principal biomechanical preparation processes, including cleaning and shaping, followed by the sterilization and obturation of the root canal, known as the endodontic triad, which is the key to a successful root canal treatment [1,2]. During this treatment, the root canal cavity must be sterilized from bacterial infection, including the opportunistic *Enterococcus*, an abnormal flora in the oral cavity habitat that generally colonizes temporarily in individuals with a weak immune system. One of these antibiotic-resistant bacteria, *Enterococcus faecalis*, has been isolated from various oral conditions, including carious lesions and chronic periodontitis, and is associated with persistent apical periodontitis [3,4]. Successful endodontic treatment depends on the selection and application of ingredients or materials for completing obturation, where the root canal is filled with synthetic materials that cover the canal and inhibit the development of the remaining microorganisms [5,6]. The root canal filling material is a combination of core material, with gutta-percha as the most widely used one and the golden standard, and a sealer, which is a paste that fills the irregular space between the walls of the gutta-percha and dentin up to the lateral channel [6,7].

An ideal sealer has many features, including being bactericidal, having the capacity to form hermetic density and adhere to the dentin wall, being easy to apply to the root canal without shrinkage after hardening (nor expanding during hardening), and having an adequate working time and consistency [7,8]. One of the most used sealer materials is mineral trioxide aggregate (MTA), a bioceramic-based material consisting of various calcium compounds such as calcium silicate (Ca_3_SiO_5_, Ca_2_SiO_4_, CaSiO_3_), calcium aluminate (CaAl_2_O_4_), and calcium sulfate (CaSO_4_·2H_2_O) [9,10]. Calcium silicate is known to have excellent broad-spectrum antibacterial power, induces apatite formation in hard tissues, and is thus osteoconductive [6,11,12]. Dicalcium silicate and tricalcium silicate produce calcium hydroxide when reacting with water, providing a strong alkalinity and absorbing carbon dioxide, which is essential for the survival of anaerobic bacteria [10,13,14]. MTA-based sealers offer advantages in their usability for teeth obturation with an open apex, perforated lesions, and resorption damage due to their ability to harden under humid conditions and induce cementogenesis and dentinogenesis with a strong bond with the dentin wall [15,16]. However, their usage is still rare because of their expensive price.

A commercial MTA generally consists of about 75 wt.% Portland cement, 20 wt.% bismuth trioxide_,_ and 5 wt.% calcium sulfate/gypsum as a setting modifier in the powder mixture [17]. Bismuth trioxide (Bi_2_O_3_) has been widely used for more than one decade as a radiopacifier in the MTA mixture [18,19,20]. Many investigations have revealed that inert Bi_2_O_3_ was not involved in the setting reaction and had no effect on Portland cement during the hydration reaction. However, the incorporation of Bi_2_O_3_ as a radiopacifier will produce flaws, increase porosity by leaving more unreacted water in the hydration reaction of the Portland cement, and may reduce the mechanical stability of the sealer [21]. Some preliminary studies showed a compositional similarity of MTA with tricalcium silicate-white Portland cement (TS-WPC), except for the absence of bismuth trioxide in the latter [22,23,24]. This compositional similarity may give it similar material characteristics that could make TS-WPC an interesting cost-effective alternative to commercial MTA. Therefore, this work aims to evaluate the potential application of a modified TS-WPC for endodontic sealer material by synthesizing and characterizing the material and evaluating its antibacterial properties.

## 2. Materials and Methods

The materials used in this work were TS-WPC powders (Indocement Ltd., Cirebon, Indonesia), bismuth trioxide (Bi_2_O_3_, Shanghai Xinglu Chemical Technology Co. Ltd., Shanghai, China), and commercial ProRoot^®^ MTA sealer package (Dentsply Sirona, Ballaigues, Switzerland). All the materials were sterilized under ultraviolet light for 1 h before use. The synthesis of modified TS-WPC (hereafter called TS-WPC for simplicity) sealer material was conducted by dissolving 80 mg of TS-WPC in 100 mL of 99.9% isopropanol, then 20 mg of Bi_2_O_3_ was mixed into the dissolved TS-WPC. The mixture was stirred for 30 min using a magnetic stirrer at a rotation speed of 400 rpm until it was homogenous, then it transferred into tubes for centrifugation at 1000 rpm for 10 min. As the supernatant was discharged, the pellet was dried under a vacuum for 60 min to evaporate the isopropanol. The TS-WPC powder samples and the ProRoot^®^ MTA (hereafter called as MTA) for comparison were then characterized by Fourier transform infrared spectroscopy (FT-IR, Spectrum 100, PerkinElmer Inc., Shelton, WA, USA) to obtain an infrared spectrum of absorption, emission, and photoconductivity for different functional groups; X-ray diffraction (XRD, D2 Phaser, Bruker Corp., Santa Barbara, CA, USA) to obtain detailed information about the crystallographic structures; and scanning electron microscopy and energy dispersive X-ray spectroscopy (SEM-EDS, JSM-6510A, Jeol Ltd., Tokyo, Japan) to observe the surface morphology at high magnification and to analyze its chemical composition.

The powder samples were also subjected to antibacterial evaluation against *E. faecalis* (ATCC 29212) using the Mueller–Hinton Agar method to determine the inhibition zone, the minimum inhibitory concentration (MIC), and the minimum bactericidal concentration (MBC). The inhibition zone was determined by spreading 100 μL of *E. faecalis* suspension onto the MHA surface in 6 mm diameter wells, then samples of 20 mg TS-WPC and MTA powders were applied onto the wells incubated for 24 h at 37 °C in an anaerobic incubator. The tests were performed in triplicate, and the inhibition zone diameter was then measured using a millimeter-scale caliper as the average of three tests. To evaluate the MIC, both samples were diluted in 9% NaCl to obtain a range of concentration up to 400,000 ppm. Then, 100 μL of each sample was inserted into the designated 96-well microplates, where 10 μL of Mueller–Hinton broth containing bacteria was then added, followed by incubation at 37 ℃ for 24 h, and ended by measuring the turbidity in a microplate reader. The MIC was indicated by wells containing the lowest concentration that was still able to inhibit bacterial growth. The MBC was evaluated by preparing two more concentrated solutions than the MIC and two more dilute solutions than the MIC, which were sub-cultured (planting) on the MHA media and incubated at 37 ℃ for 24 h. This planting was performed to overcome the microplate reader’s ability to read the plate’s dilution turbidity by measuring its optical density alone, without detecting the separate turbidity caused by the two sample materials’ different densities. The MBC was determined from the agar media with no bacterial colony.

## 3. Results

The TS-WPC and MTA samples presented some similarities in their IR spectra before and after hydration (Figure 1a). Hydration was performed by reacting the samples with distilled water to simulate conditions during which a sealer is applied to a root canal. Table 1 details the association of each peak with its corresponding functional group and chemical bond. Before hydration, both the samples showed strong peaks at 883.67 cm^−1^ (TS-WPC) and 876.40 cm^−1^ (MTA), associated with the Si-C/C=C bond of the alkenes group. Weaker peaks were identified as Si-O bonds at 1144.89 and 1153.90 cm^−1^ and as C-H/C-O bonds (alkanes/alkanols) at 1486.18 and 1456.61 cm^−1^ for each TS-WPC and MTA sample, respectively. Beyond the fingerprint band, peaks associated with the Si-H bond occurred at 2359 cm^−1^ for both samples. After hydration, a peak appeared at 3395 cm^−1^ on both samples’ IR spectra, indicating the presence of hydroxide compounds such as Ca(OH)_2_. The alkenes group’s presence was stronger in both samples, as indicated by the high intensity peaks of C-H/C-O at 1415 cm^−1^ and Si-C/C=C at 871 cm^−1^.

Further characterization by XRD identified the presence of four compounds in both samples before hydration (Figure 1b), which are tricalcium silicate (Ca_3_SiO_5_), bismuth trioxide (Bi_2_O_3_), dicalcium silicate (Ca_2_SiO_4_), and tricalcium aluminate (Ca_3_Al_2_O_6_). After hydration, the four compounds’ peaks were still present with the new appearance of calcium hydroxide (Ca(OH)_2_) peaks. The peaks appeared very weak due to the relatively short contact time (30 min) between the samples and water during hydration. Both the XRD and FT-IR results confirmed a chemical similarity between the TS-WPC and MTA samples, while SEM observation showed their morphological similarity (Figure 2). Both powders had an octahedral shape, with an average size of 5 μm, and elemental bismuth was present on the powders as Bi_2_O_3_.

The bacterial inhibition zones of the two tested samples indicated a similar measured diameter (Figure 3). The average diameter of the formed clear area was 15.33 mm for the TS-WPC and 15.36 mm for the commercial MTA. Table 2 shows that some *E. faecalis* colonies still existed at the concentration of 12,500 ppm. The bacteria were killed and declared to be depleted at the concentration of 25,000 ppm.

## 4. Discussion

The TS-WPC powder samples were characterized using FT-IR, XRD, and SEM-EDS, along with samples of the MTA as a comparison. The characterization results showed some similarities between the lab-made TS-WPC samples and the commercial MTA sealer in terms of chemistry, crystallinity, and granulometry. On the IR spectra, both samples presented strong peaks of the Si-C/C=C bond, weaker peaks of the Si-O and C-H/C-O bonds, and beyond the fingerprint band peaks of the Si-H bond. Once hydrated, new peaks of Ca(OH)_2_ appeared while preserving the high intensity peaks of C-H/C-O and Si-C/C=C bonds. The appearance of Ca(OH)_2_ was again identified on the XRD patterns of hydrated samples, while before hydration four main compounds were identified as Ca_3_SiO_5_, Bi_2_O_3_, Ca_2_SiO_4_, and Ca_3_Al_2_O_6_. These results were in line with the previous research that confirmed the presence of Ca(OH)_2_ [25]. Calcium hydroxide (Ca(OH)_2_) has long been used as one of the most effective root canal medicaments. Calcium hydroxide has a working action by releasing Ca^2^⁺ ions which play a role in the mineralization process of tissues and OH⁻ ions, which can provide an antimicrobial effect by increasing the pH so that an alkaline environment is formed, which causes most of the microorganisms in the root canals to be unable to survive [26]. The characterization results clearly showed that TS-WPC with added 20 wt.% Bi_2_O_3_ as a radiopacifier had the same configuration of peaks as that of the commercial MTA, and both materials were compositionally similar. These similar material characteristics should provide similar properties—i.e., antibacterial capabilities. In terms of granulometry, both samples have octahedral shape powders with an average size of 5 μm, and elemental bismuth was evidently present on the powders as Bi_2_O_3_, as observed in the work of Bedoya-Hincapie et al. [27].

The usage of endodontic sealers’ to perform root canal fillings in obturation procedures is an established mainstay in endodontics and plays a crucial role in the success of the treatment. Therefore, these materials should exhibit a set of characteristics that allow successful root canal filling with the resolution of periapical inflammatory and/or infectious processes and prevent further microbial contamination [28]. Based on the results of the antibacterial evaluation against *E. faecalis* using the Mueller–Hinton Agar method, the two samples presented similar diameters of bacterial inhibition zones of around 15 mm, falling within the classification of intermediate sensitivity, with a diameter between 14 and 22 mm [29]. The consistent dense sealer after setting time within 24 h is one of the limiting factors for the diffusion process of antibacterial agent to the further agar area. Therefore, this TS-WPC sealer’s application will require direct contact in the root canal cavity to eliminate bacteria optimally [29,30,31]. The TS-WPC sample achieved its minimum inhibitory concentration (MIC) at 12,500 ppm, while its minimum bactericidal concentration (MBC) was reached at 25,000 ppm. This MBC value is interestingly far below the concentration commonly used in root canal sealer applications.

The antibacterial capability of both samples can be attributed to the presence of a Ca(OH)_2_ compound that created a strong alkaline condition and provided hydroxyl ions, which was consistent with the other finding that the application of intracanal medicaments containing Ca(OH)_2_ compounds was a competent of forming negative cultures of *E. faecalis* [32,33]. Most pathogenic bacteria in the root canal cannot survive the strong alkaline conditions, with a pH of around 12.5, produced by Ca(OH)_2_. Generally, bacteria will be eliminated sometime after contact with Ca(OH)_2_, as they can only tolerate a pH range between 6 and 9 [34]. The high pH conditions created by Ca(OH)_2_ inhibit enzyme activity during bacterial cell metabolism [35]. Alkalinity induces the breakdown of ionic bonds that maintain the tertiary structure of proteins, thus only covalent bonds are maintained leading to the formation of an irregular bond of polypeptide chains. These changes eventually cause the loss of the enzymes’ biological activity, thus damaging the cell metabolism [30].

The second antibacterial capability of Ca(OH)_2_ comes from the hydroxyl ions, which are high free radical oxidants with strong reactivity toward bacterial cytoplasmic membrane [31]. Bacterial cytoplasmic membranes have essential functions in cell defense mechanisms, such as selective permeability and fluid transport; secretion from the hydrolysis exoenzyme; and delivering enzymes and molecules that function in DNA biosynthesis, cell wall polymers, and membrane lipids [30]. Hydroxyl ions induce lipid peroxidation, which damages phospholipids and the cell membrane structure. Hydroxyl ions avert hydrogen atoms from unsaturated fatty acids in producing free radicals from lipids. These free radicals react with oxygen and form peroxide lipid radicals that avert other hydrogen atoms from secondary fatty acids to re-produce other lipid peroxides. Peroxide acts as a free radical that automatically initiates the catalysis chain reaction, losing more fatty acids and sustaining more severe membrane damage. Hydroxyl ions also react with bacterial DNA and induce the division of DNA from the chain structure unity that can cause the loss of bacterial genetic code, leading to a failure in DNA replication, and thus cell activity does not occur [30]. Another contributing factor to the antibacterial capability of Ca(OH)_2_ is its ability to absorb CO_2_ stored in the periradicular tissue of the root canal cavity, eliminating the essential atmosphere for anaerobic bacterial survival of bacteria such as *E. faecalis* [34,35].

This modified TS-WPC could become a cost-effective alternative endodontic sealer material once its biocompatibility is confirmed. It may become one among many alternative therapies for treating root canal problems including the easily accessible stem cells that can be isolated from various tissues of the oral cavity and become sources for bone and dental regeneration [36].

## 5. Conclusions

A modified tricalcium silicate-white Portland cement (TS-WPC) was synthesized using a simple solution method and possessed similar material characteristics and antibacterial properties to those of a commercial mineral trioxide aggregate (MTA)-based sealer material. The TS-WPC was characterized by powder morphology, hydroxyl and silicate groups, and bismuth trioxide and tricalcium silicate phases like those of a commercial MTA-based sealer. The TS-WPC and MTA samples exhibited antibacterial activity against the anaerobic *E. faecalis* bacteria, with minimum inhibitory and killing concentrations of 12,500 ppm and 25,000 ppm, respectively. Although both materials’ TS-WPC and the commercial MTA were proven to be compositionally similar, further studies of TS-WPC are awaited based on these findings to improve its mechanical stability and biocompatibility. The modified TS-WPC has the potential to become a cost-effective alternative endodontic sealer material.

## Figures and Tables

**Figure 1 materials-14-00417-f001:**
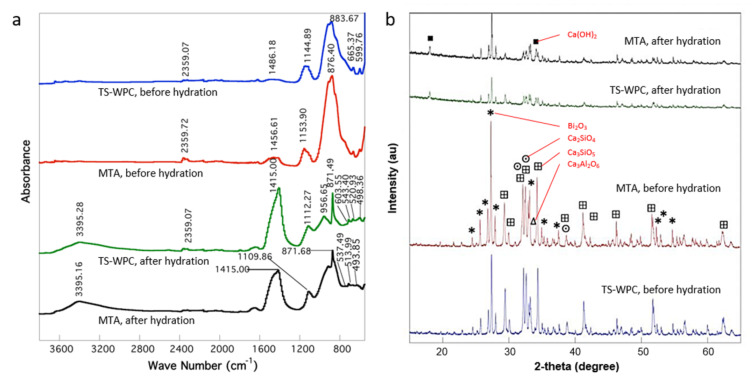
(**a**) FT-IR spectra and (**b**) XRD pattern results of the tricalcium silicate-white Portland cement (TS-WPC) and the commercial mineral trioxide aggregate (MTA) samples before and after hydration.

**Figure 2 materials-14-00417-f002:**
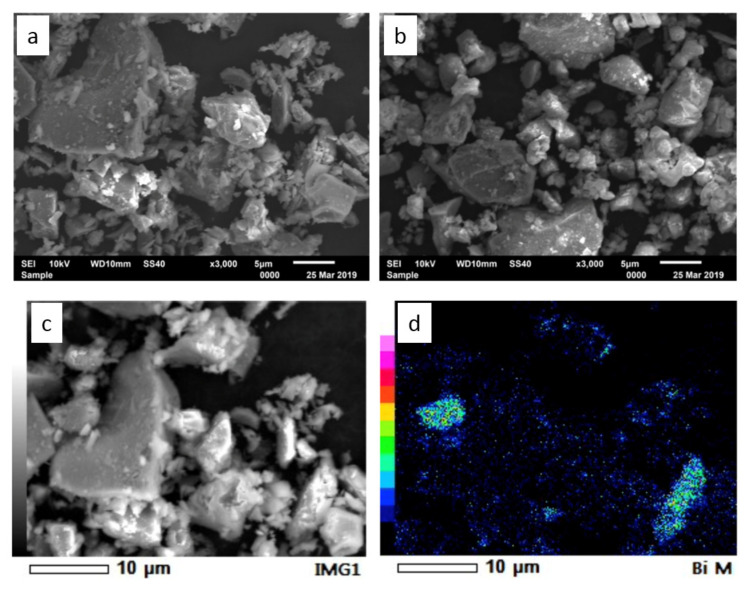
Secondary electron micrographs of (**a**) TS-WPC and (**b**) MTA samples, (**c**) backscattered electron micrograph of TS-WPC, (**d**) elemental mapping of bismuth on the TS-WPC sample. All the sample pictures are from before hydration.

**Figure 3 materials-14-00417-f003:**
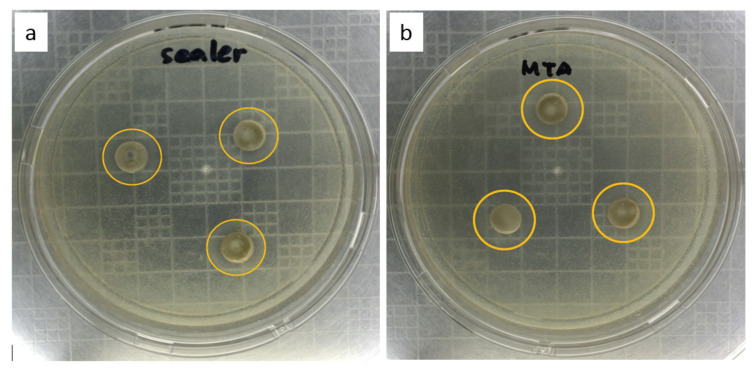
Bacterial inhibition zone (yellow circle) of (**a**) TS-WPC and (**b**) MTA samples against *E. faecalis.*

**Table 1 materials-14-00417-t001:** FT-IR peak identification.

TS-WPC		MTA	
Wavenumber (cm^−1^)	Bond/Type of Functional Group	Wavenumber (cm^−1^)	Bond/Type of Functional Group
Before hydration			
599.75	C-H (alkanes)		
665.37			
883.67	Si-C/C=C (alkenes)	876.40	Si-C/C=C (alkenes)
1144.89	Si-O	1153.90	Si-O
1486.18	C-H (alkanes)/C-O (alkanols)	1456.61	C-H (alkanes)/C-O (alkanols)
2359.07	Si-H	2359.72	Si-H
After hydration			
451.71	Aromatic group (S-O)/sulfate ionic bonds (SO_4_)^−^	436.77	Aromatic group (S-O)/sulfate ionic bonds (SO_4_)^−^
467.69		445.38	
498.36		466.56	
		475.90	
		493.85	
520.93	Fingerprint band/Si-O (bending)/Ca=O	513.99	Fingerprint band/Si-O (bending)/Ca=O
543.40		537.49	
603.55	Fingerprint band		
871.49	Si-C/C=C (alkenes)	871.68	Si-C/C=C (alkenes)
956.65	Si-O (stretching)		
1112.27	Si-O	1109.86	Si-O
1408.59	C-H (alkanes)/C-O (alkanols)	1415.00	C-H (alkanes)/C-O (alkanols)
2359.07	Si-H		
3395.28	O-H	3395.16	O-H

**Table 2 materials-14-00417-t002:** Minimum bactericidal concentration of the TS-WPC samples.

Observation							12,500 ppm	25,000 ppm	50,000 ppm	100,000 ppm	200,000 ppm	400,000 ppm	Notes
No.	1	2	3	4	5	6	7	8	9	10	11	12	
A	−	−	+	+	+	+	+	−	−	−	−	−	
B	−	−	+	+	+	+	+	−	−	−	−	−	
C	−	−	+	+	+	−	−	−	−	−	−	−	
D	−	−	−	−	−	−	−	−	−	−	−	−	*
E	−	−	+	+	+	+	−	−	−	−	−	−	
F	−	−	+	+	+	+	+	−	−	−	−	−	
G	−	−	+	+	+	+	+	−	−	−	−	−	
H	−	−	−	−	−	−	−	−	−	−	−	−	**
	***												

Notes: * Media mixed with MTA with no bacteria; ** media mixed with TS-WPC, with no bacteria; *** control media without testing materials and bacteria (100 mL of MHB). Letters A-H and numbers 1–12 indicate the tested plates and sealer concentrations, respectively. A negative sign (−) means the bacterial colony died, positive sign (+) means the bacterial colony survived.

## Data Availability

No new data were created or analyzed in this study. Data sharing is not applicable to this article.

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
