# Peer review of "Synthesis, Characterization, and Antibacterial Evaluation of a Cost-Effective Endodontic Sealer Based on Tricalcium Silicate-White Portland Cement"

_materials, 2021, doi:10.3390/ma14020417_

Round 1

Reviewer 1 Report

Language needs improvement, especially in the Introduction and the Discussion. One of the major concerns here is the English. The authors should have the manuscript looked at for language and sentence composition. There are a lot of sentences in the manuscript that do not make sense because of the English.

Abstract: the authors are suggested to provide a bit more detailed information in method and result parts.

Introduction:

Whole of discussion is poorly written and insufficient. The discussion is also misleading

STATISTICAL

Did t-test alone have been performed for each experiment ? could you be more specific ?

Relevant parameters are not included.

Reviewer 2 Report

Dear Authors,

I read your manuscript, and it can be considered interesting.

The manuscript presents some leaks in its structure, among them:

Material and methods

Line 81 and line 82 … powder samples were then characterized by using FT-IR (Spectrum 100, PerkinElmer Inc., 82 Shelton, USA), XRD (D2 Phaser, Bruker Corp., Massachusetts, USA), and SEM-EDS.

What is FT-IR, XRD and SEM-EDS?

Results

Line  81 and 82. “WPC powder samples were then characterized by using FT-IR (Spectrum 100, PerkinElmer Inc., Shelton, USA), XRD (D2 Phaser, Bruker Corp., Massachusetts, USA), and SEM-EDS.

I advice this paragraph in the discussion and rephrase al results.

Line 158. What are the similar material characteristics that should render similar properties?

Please re-write the discussion section to make it more appealing.

As biocompatibility is one of the main properties of root canal sealers, as these materials come into direct contact with periradicular tissues, it is importante to refer this property and cite this systematic review:

Fonseca DA, Paula AB, Marto CM, Coelho A, Paulo S, Martinho JP, Carrilho E, Ferreira MM. Biocompatibility of Root Canal Sealers: A Systematic Review of In Vitro and In Vivo Studies. Materials (Basel). 2019 Dec 9;12(24):4113. doi: 10.3390/ma12244113. PMID: 31818038; PMCID: PMC6947586.

Best regards

Reviewer 3 Report

The association of Portland cement and bismuth oxide is very interesting, however, there are several studies in the literature that proposed this mixture, and they should be discussed in the text.

The conclusions should emphasize that the new mixture showed similar results, respect MTA, for the analysis performed in this specific study, but other experimentations are needed to verify if the biocompatibility and other features of the novel compound are overlapping those of MTA.

Round 2

Reviewer 1 Report

Dear authors,

Thank you for improving the manuscript.

However, in this phrase, the references are no correct.
"The MTA-based sealers offer advantages in their usability for teeth obturation with an open apex, perforated lesions, and resorption damages, due to their ability to harden under humid conditions and induce cementogenesis and dentinogenesis with a strong bond with the dentin wall [9,15-18]".
I recommend adding to the following references:
Rodríguez-Lozano FJ, López-García S, García-Bernal D, Tomás-Catalá CJ, Santos JM, Llena C, Lozano A, Murcia L, Forner L. Chemical composition and bioactivity potential of the new Endosequence BC Sealer formulation HiFlow. Int Endod J. 2020 Sep;53(9):1216-1228. doi: 10.1111/iej.13327. Epub 2020 Jun 18. PMID: 32412113.

Güven EP, Taşlı PN, Yalvac ME, Sofiev N, Kayahan MB, Sahin F. In vitro comparison of induction capacity and biomineralization ability of mineral trioxide aggregate and a bioceramic root canal sealer. Int Endod J. 2013 Dec;46(12):1173-82. doi: 10.1111/iej.12115. Epub 2013 Apr 26. PMID: 23617276.  
